🔓| Open Peer Review | Genomics and Proteomics | Research Article

# Characterization of RcgA and RcgR, two rhizobial proteins involved in the modulation of plasmid transfer

Lucas G. Castellani,[1] María Delfina Cabrera,[1] Abril Luchetti,[1] Juliet F. Nilsson,[1] Julieta Pérez-Giménez,[1] Luis Alfredo Bañuelos-Vazquez,[2] Alma Alva,[3] Daniel Wibberg,[4] Tobias Busche,[4] Jörn Kalinowski,[4] Andreas Schlüter,[4] Alfred Pühler,[4] Karsten Niehaus,[4] Mariano Pistorio,[1] Gonzalo Torres Tejerizo[1]

**ABSTRACT** Plasmid conjugative transfer (CT) is a major mechanism of horizontal gene transfer in bacteria, facilitating genome evolution and dissemination of adaptive traits. Due to the energetic cost of CT, its regulation becomes an important process to ensure energetic balance within cells. In *Rhizobium favelukesii*, the plasmid pLPU83a belongs to group I-C of rhizobial plasmids, which require the transcriptional regulator TraR for CT. In well-characterized systems, TraR typically activates conjugative genes in response to quorum-sensing (QS) signals such as acyl-homoserine lactones. However, pLPU83a does not respond to these signals, raising questions about how TraR is regulated in this system. This study addresses the function of RcgA and RcgR, two proteins encoded upstream of *traR* on pLPU83a, whose function has previously been associated with CT modulation. Through proteomic, transcriptomic, and microscopy approaches, we show that RcgR acts as a repressor of CT, inhibiting *traR* expression and, therefore, the transcription of genes involved in CT, thereby reducing plasmid transfer rate. In contrast, RcgA is essential for CT but does not affect the expression of CT genes; it is localized at the membrane and may play a structural role in the mating pair formation system. Functional assays revealed that the repression facilitated by RcgR is independent of the anti-activator TraM and that TraR is essential for transfer even in the absence of RcgR. These findings locate RcgA and RcgR as key elements of a new circuit that modulates rhizobial plasmid conjugation and propose a novel mechanism of TraR control in systems uncoupled from QS signaling.

**IMPORTANCE** Plasmid transfer is a central mechanism of gene exchange in bacteria, enabling the spread of traits with ecological and evolutionary relevance. *Rhizobium favelukesii* is a soil bacterium that carries multiple plasmids, including pLPU83a, which serves as a model to study conjugative transfer. This plasmid requires the transcriptional regulator TraR for transfer but—unlike classical systems—lacks the cognate gene that encodes the AHL synthase typically involved in quorum-sensing regulation. In previous work, two novel proteins encoded on pLPU83a, RcgA and RcgR, were identified as key elements in this regulatory system. Here, we further characterized their roles: RcgR represses the transcription of *traR* and, consequently, that of all conjugative genes, while RcgA is essential for transfer and localizes to the membrane, suggesting a structural function. These results provide mechanistic insight into how plasmid transfer is regulated in systems uncoupled from quorum sensing, highlighting alternative layers of control in bacterial conjugation.

**KEYWORDS** rhizobia, plasmid, conjugation, RNAseq, proteomics

**Peer Reviewer** Elisabeth Grohmann, Beuth Hochschule für Technik, Berlin, Germany

Address correspondence to Gonzalo Torres Tejerizo, gatt@biol.unlp.edu.ar.

Lucas G. Castellani and María Delfina Cabrera contributed equally to this article. Author order was determined by drawing straws.

The authors declare no conflict of interest.

See the funding table on p. 15.

Plasmids are widely distributed extrachromosomal elements that replicate using their own mechanisms and can be transferred between microorganisms (1). These

characteristics make plasmids one of the most important factors contributing to the evolution of bacterial genomes and the dissemination of relevant traits such as antibiotic resistance genes and pathogenicity or symbiotic determinants (2–4).

Rhizobia are a group of soil bacteria able to perform symbiotic nitrogen fixation (SNF) in association with the roots of leguminous plants (5, 6). The necessary traits for SNF are usually encoded in plasmids, called symbiotic plasmids or pSyms. Rhizobia often harbor other plasmids in addition to pSyms, called cryptic or accessory plasmids, which are involved in other relevant functions (7). Evidence of horizontal gene transfer (HGT) events suggests that the evolution of rhizobia has been achieved by plasmid acquisition (8, 9). Due to their ability for nitrogen fixation, the potential use as inoculants makes these organisms a very important focus of study in agronomy. Even though the search for microorganisms with a higher performance on nitrogen fixation is important, it is also necessary to study the behavior of the plasmids present in these microorganisms in order to understand how these elements could disseminate among soil bacterial populations (10–13).

In general, for a plasmid to be transferred by conjugation, two different systems are required: a DNA transfer and replication system (Dtr) and a mating pair formation system (Mpf) (14). Dtr is composed of proteins involved in the recognition of the origin of transfer (oriT) and DNA processing (referred to as Tra proteins). Mpf, in turn, consists of proteins involved in the formation of a type 4 secretion system (T4SS) through which DNA is transferred to recipient cells (referred to as Trb or VirB proteins). Regarding process regulation, two main mechanisms are well described in rhizobia. One of them relies on the rctA/rctB genes, where rctA encodes a protein with a structure related to transcriptional regulators with a winged-helix DNA‑binding domain. This structure allows RctA to bind to the promoter region of the virB operon (or trb), interfering with the expression of Mpf genes. The gene rctB encodes a protein that represses RctA function. Accordingly, an overexpression of RctB enhances conjugative plasmid transfer. Nevertheless, the signal molecule that triggers the regulatory cascade that activates rctB transcription is not known (7, 15–17). The other well-described regulatory system is based on TraR, a LuxR-like transcriptional regulator. LuxR is the archetypal quorum-sensing (QS)-responsive transcriptional regulator. In known systems, TraR activates conjugative genes in response to acyl-homoserine lactone (AHL) concentrations, a small diffusible molecule produced by TraI (an AHL-synthase in rhizobial plasmids), generally encoded in the same plasmid near traR. As a result, plasmid transfer responds to a QS mechanism, increasing when AHL concentrations are higher (18, 19). In these TraR systems, traM plays an important role as it encodes a TraR anti-activator protein involved in a fine-tuning process regulation (20–22). Although the main components of these systems have been described, genes encoding hypothetical proteins were linked to rhizobial conjugative transfer in recent years, demonstrating that other systems could arise from the study of these proteins (23–29). Furthermore, new actors could be involved in the modulation of previously described processes.

Rhizobial plasmids carrying the master regulator traR gene among conjugative genes were classified into four groups, according to the organization of conjugative clusters (30). TraR of plasmids belonging to groups I-A and I-B responds to AHL concentration, while plasmids of groups I-C and I-D do not seem to respond to population density. In particular, plasmids of group I-C do not possess an AHL synthase gene (traI) in the conjugative region, while the traR present in the region is essential for CT (25, 30, 31). This raises the question of which other actors are involved in the CT regulation of these plasmids and what their relation is with TraR. The plasmid pRfaLPU83a (hereafter pLPU83a) from Rhizobium favelukesii LPU83 is a model plasmid belonging to group I-C. It has been described that TraR is essential for pLPU83a transfer (31), even though it does not respond to either AHL or population density (30). Recently, we have identified RcgA and RcgR, two hypothetical proteins encoded by pLPU83a and several other plasmids that belong to group I-C. The corresponding genes are located in tandem between Dtr and Mpf genes, adjacent to traR, and are involved in the modulation of

pLPU83a conjugative transfer. For RcgA, it was demonstrated that it is essential for plasmid transfer, while RcgR acts as a repressor, reducing, at least, *traR* expression. Strikingly, *rcgA* and *rcgR* were also found in a group of plasmids where *traR* is not present, although its characterization has not been carried out (28). With the aim of improving the knowledge of the conjugative system operating in plasmids belonging to group I-C, and the mechanism by which RcgA and RcgR modulate CT, we characterized RcgA and RcgR by proteomic, transcriptomic, microscopy, and molecular biology approaches. In addition to the characterization of both proteins, this study shows new putative targets that could be involved in conjugative plasmid transfer regulation.

## RESULTS

### Cellular localization of RcgA and RcgR

The main genes involved in CT are located in the Mpf and Dtr regions. In the model plasmid pLPU83a, *rcgA* and *rcgR* are located between these two regions. Even though *rcgA* and *rcgR* form an operon, they modulate conjugation in different manners, generating opposite phenotypes in corresponding mutant strains: a Δ*rcgR* mutant shows a higher conjugative frequency of pLPU83a, while a Δ*rcgA* mutant is not able to perform CT (28). As transmembrane domains were predicted in RcgA, it was hypothesized that it could be a protein located at the membrane, while RcgR seemed to be a cytoplasmic protein (28). To get a deeper insight into the possible role of these proteins, translational fusions of each one to GFP were constructed, aiming to determine their cellular localization. Once the translational fusions were obtained, *rcgA* and *rcgR* mutant strains were complemented with the corresponding GFP fusion to assess the functionality of each protein. Both fusions were functional since they were able to restore the wild-type conjugative phenotype. As a first approach to evaluate localization, a classical protein isolation procedure was carried out in order to obtain cytoplasmic and membrane protein fractions for each strain [*R. favelukesii* LPU83-13Δ*rcgA* (pBBR1MCS-5::*rcgA*::GFP) and *R. favelukesii* LPU83-13Δ*rcgR* (pBBR1MCS-5::*rcgR*::GFP)], and a Western blot against GFP was performed. As a control, proteins obtained from a strain expressing only GFP were analyzed [*R. favelukesii* LPU83-13 (pBBR1MCS-5::GFP)]. GFP (ca. 27 kDa) was detected in both fractions in its expected size along with a second band with low intensity of ca. 17 kDa (see Fig. S1 at https://doi.org/10.35537/10915/189658). A similar fact was observed by other authors, and it was hypothesized that these are non-specific bands (32). While RcgR::GFP was found in both cytoplasmic and membrane protein fractions in the expected size (ca. 65 KDa), RcgA::GFP was detected, as predicted, exclusively in the membrane protein fraction. However, its apparent molecular weight was higher than anticipated (~90 kDa), migrating above the highest molecular weight marker (180 kDa). This higher-than-expected size may reflect the anomalous migration often reported for transmembrane proteins, which can bind SDS irregularly (33, 34), or the possibility that RcgA::GFP forms SDS-resistant dimers (35).

In order to analyze the cellular localization of the proteins, the same strains were used for fluorescence microscopy, and the bacterial membrane was labeled with a specific probe (FM 4-64 Dye). To overcome the resolution limit of conventional fluorescence microscopy and accurately determine protein localization, image resolution was enhanced by a factor of 10 using the Super-Resolution Radial Fluctuation (SRRF) computational algorithm (36). GFP was employed as a control, which was observed lying scattered in the cytoplasm (Fig. 1A through C). For RcgR, it was not possible to determine a specific localization as images showed accumulation at the cell poles, distribution across the cytosol, or co-localization with the membrane dye (Fig. 1D through F). Results for RcgA were more conclusive, since all the images showed a co-localization between this protein and the membrane (Fig. 1G through I). To quantify our results, an analysis of the fluorescence intensity distribution along the cells was performed (see Fig. S2 at https://doi.org/10.35537/10915/189658). This analysis confirmed the cytosolic distribution of free GFP, showing high GFP signals throughout the whole cell (from one pole, position 1, to the other one, position −1) (see Fig. S2A at https://doi.org/

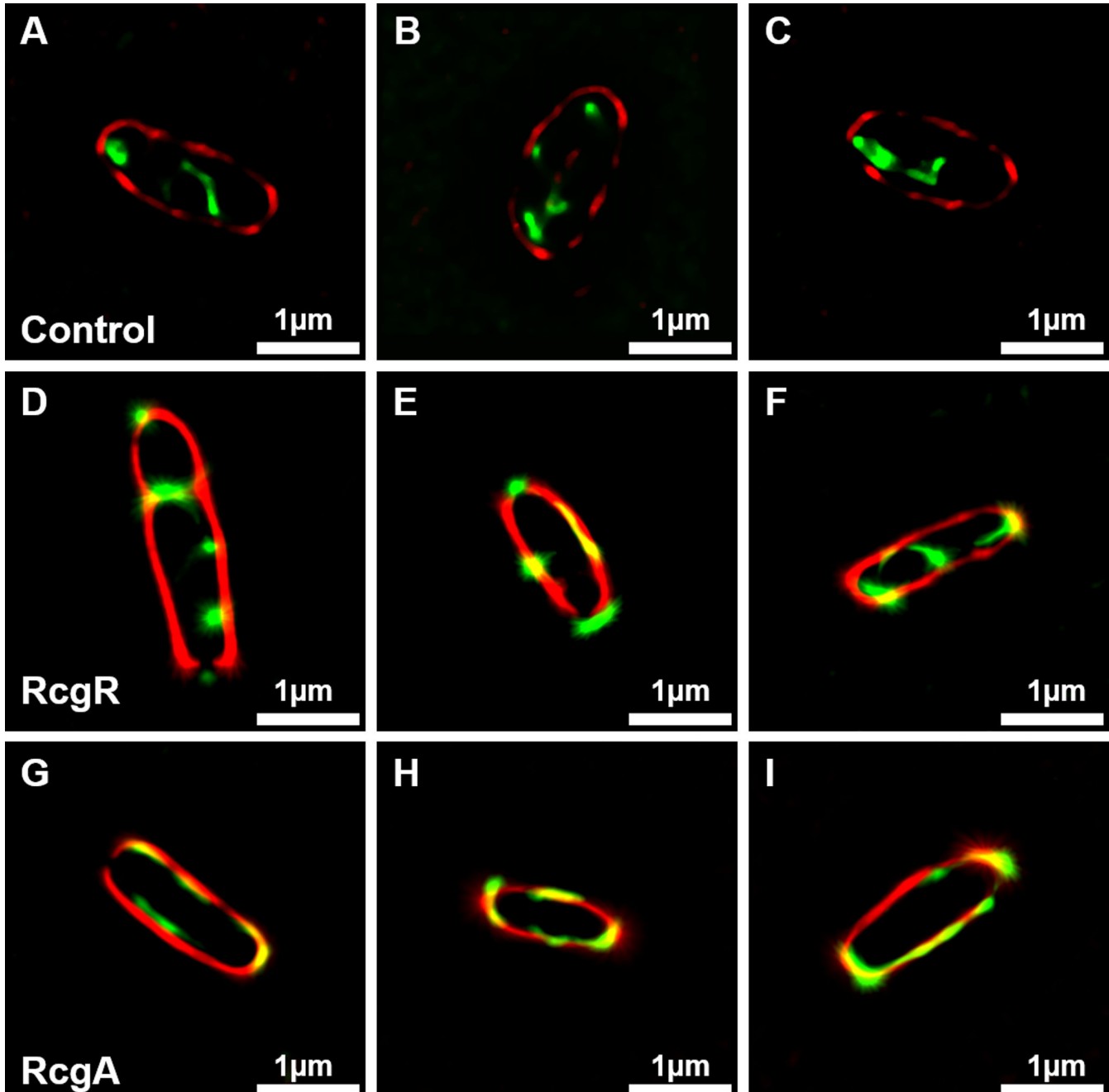

**FIG 1** Cellular localization of RcgR and RcgA by high-resolution fluorescence microscopy. Representative enhanced resolution images of *R. favelukesii* expressing GFP in the wild-type strain (A–C), RcgR::GFP in Δ*rcgR* background (D–F), and RcgA::GFP in Δ*rcgA* background (G–I). All strains were labeled with FM 4‑64 Dye. Red color represents the fluorescent emission of FM 4-64, green color represents the emission of GFP fluorescent protein, and yellow color represents colocalization of FM 4-64 and GFP fluorescent signals. The scale bar is at the bottom right of each image.

10.35537/10915/189658). RcgR showed GFP intensity peaks along with those of the membrane probe at the cell poles but also exhibited GFP signals between them (see Fig. S2B at https://doi.org/10.35537/10915/189658). In agreement with our Western blot assays, RcgA showed the highest GFP intensities near those of the membrane probe, confirming its membrane localization in most of the analyzed cells (see Fig. S2C at https://doi.org/10.35537/10915/189658).

## Stability assays of *rcgA* and *rcgR* mutants

Several studies have found that proteins involved in plasmid stability affect plasmid transfer (37–39). A particular case was the system described in plasmid R388 by Guynet et al. (37), where the *stbABC* operon regulates the conjugation transfer rate and partition. Based on these data, we wondered if *rcgAR* could affect conjugative transfer by altering plasmid stability. To address this question, stability assays of both plasmid mutants were carried out. For this, plasmids pLPU83aΔ*rcgR,* pLPU83aΔ*rcgA*, and a double mutant pLPU83aΔ*rcgA*Δ*rcgR* were grown for 40 generations, and the proportion of plasmid-lacking CFU was evaluated as described in Materials and Methods. These results showed that plasmid stability was not affected since all three evaluated plasmids (pLPU83aΔ*rcgR,* pLPU83aΔ*rcgA*, and pLPU83aΔ*rcgA*Δ*rcgR*) were conserved among the population (see Fig. S3 at https://doi.org/10.35537/10915/189658).

## Proteomic analysis of *rcgA* and *rcgR* mutants

Previous whole-cell proteomic analysis by the data-dependent acquisition method of *R. favelukesii* LPU83-13 and the *rcgA* and *rcgR* mutants showed low abundance for proteins belonging to Dtr and Mpf of pLPU83a (28). Nevertheless, it was possible to observe an overexpression of TraR and TraC (Dtr proteins) in the *rcgR* mutant. In order to improve the proteomic characterization of both mutants, we performed a new whole-cell proteomic analysis, using the data-independent acquisition method (see Materials and Methods), which allows a deeper analysis of peptide abundance (40). This new analysis confirmed the overexpression of TraR and other conjugative proteins in the *rcgR* mutant, since they were detected in the mutant but not in the wild-type strain (see Table S3 at https://doi.org/10.35537/10915/189658, sheet "Δ*rcgR* ON"). On the other hand, some proteins were present in the wild-type strain but not in the Δ*rcgR* strain (see Table S3 at https://doi.org/10.35537/10915/189658, sheet "Δ*rcgR* OFF"). However, none of these proteins are related to the conjugative process nor encoded in pLPU83a. When On/Off proteins were analyzed in the Δ*rcgA* strain, only three proteins were present or absent (see Table S3 at https://doi.org/10.35537/10915/189658, sheets "Δ*rcgA* ON" and "Δ*rcgA* OFF"), although their function does not seem to be related to plasmid CT.

Considering that some proteins belonging to Mpf can be found as extracellular proteins (41–43), an analysis of the exo-proteomic fraction of each mutant and the wild-type strain was also performed. Consistently, conjugation-related proteins were detected in the extracellular fraction of the *rcgR* mutant (see Table S4 at https://doi.org/10.35537/10915/189658, sheets "*R. fav*. LPU83-13 Δ*rcgR* ON" and "*R. fav*. LPU83-13 Δ*rcgR* DEP"). Regarding the mutation in *rcgA*, the peptides with differential abundance found were not apparently related to CT (see Table S4 at https://doi.org/10.35537/10915/189658, sheets "*R. fav*. LPU83-13 Δ*rcgA* ON" and "*R. fav*. LPU83-13 Δ*rcgA* OFF").

## RNAseq analysis of *rcgA* and *rcgR* mutants

The proteomic approach suggests that RcgA and RcgR do not act together during their CT-related functions. Although both mutants have proteins with differential expression when compared to the wild-type strain, those with functions related to plasmid CT were only found in the *rcgR* mutant. In order to obtain a deeper knowledge of the regulatory circuit involving RcgR and aiming to decipher if RcgA is directly involved in other regulatory networks, a transcriptomic analysis of both mutants was performed. For this, RNA samples from both mutants and the wild-type strain were obtained and prepared for RNA sequencing. For each strain, CT frequencies were determined in parallel with cell harvesting in order to confirm that the phenotypes were the same as we described previously (28). A list of differentially expressed genes ($P$-adjusted value < 0.05 and Log$_2$ Fold Change < −1 or Log$_2$ Fold Change > 1) in the *rcgR* mutant is shown in Table 1. The differential expression analysis revealed that RcgR regulates specifically CT-related genes and a few more hypothetical genes. The mutant in *rcgR*

**TABLE 1** Differentially expressed genes in LPU83-13Δ*rcgR* strain in comparison to LPU83-13

| Locus tag | Gene product/putative function | Log$_2$ fold change | *P*-adj. value |
|---|---|---|---|
| LPU83_pLPU83a_0119 | DUF3991 domain-containing hypothetical protein | 2.76 | 2E-18 |
| LPU83_pLPU83a_0136 | Putative thermonuclease | 1.10 | 2E-03 |
| LPU83_pLPU83a_0137 | WGR domain-containing hypothetical protein | 1.71 | 1E-17 |
| LPU83_pLPU83a_0138 | TraG | 3.98 | 1E-82 |
| LPU83_pLPU83a_0139 | TraD | 5.15 | 5E-35 |
| LPU83_pLPU83a_0140 | TraC | 4.50 | 2E-61 |
| LPU83_pLPU83a_0141 | TraA | 4.32 | 5E-138 |
| LPU83_pLPU83a_0142 | TraF | 4.54 | 5E-10 |
| LPU83_pLPU83a_0143 | TraB | 3.56 | 6E-45 |
| LPU83_pLPU83a_0144 | TraH | 3.27 | 6E-44 |
| LPU83_pLPU83a_0148 | RcgA | −1.13 | 2E-04 |
| LPU83_pLPU83a_0149 | TraR | 7.79 | 0E+00 |
| LPU83_pLPU83a_0151 | TrbI | 3.89 | 9E-138 |
| LPU83_pLPU83a_0152 | TrbH | 4.54 | 2E-80 |
| LPU83_pLPU83a_0153 | TrbG | 4.75 | 5E-112 |
| LPU83_pLPU83a_0154 | TrbF | 5.32 | 2E-128 |
| LPU83_pLPU83a_0155 | TrbL | 5.24 | 3E-155 |
| LPU83_pLPU83a_0156 | TrbK | 5.50 | 5E-63 |
| LPU83_pLPU83a_0157 | TrbJ | 5.28 | 0E+00 |
| LPU83_pLPU83a_0158 | TrbE | 5.77 | 0E+00 |
| LPU83_pLPU83a_0159 | TrbD | 6.25 | 5E-96 |
| LPU83_pLPU83a_0160 | TrbC | 5.91 | 2E-94 |
| LPU83_pLPU83a_0161 | TrbB | 6.72 | 2E-211 |

showed a remarkable increase in *traR* levels, with the gene exhibiting the highest fold change (Log$_2$ Fold Change = 7.79). This result is coherent with a previously evaluated transcriptional fusion of *traR* (28). In addition, all genes belonging to Dtr (*tra* genes) and Mpf (*trb* genes) of pLPU83a were found to be overexpressed, confirming the observed phenotype of higher conjugative transfer frequencies and the proteomic results. Three more genes present in pLPU83a were overexpressed, two of them encoding hypothetical proteins (LPU83_pLPU83a_0119 and LPU83_pLPU83a_0137) and the other one encoding a putative thermonuclease (LPU83_pLPU83a_0136). Remarkably, two of these genes (LPU83_pLPU83a_0136 and LPU83_pLPU83a_0137) are located downstream of *traG* (coupling protein [CP], encoded in the Dtr region). Hence, its proximity to this system could imply a function related to the conjugative process. Figure 2 shows the fold change and the organization of the Dtr and Mpf genes, including LPU83_pLPU83a_0136 and LPU83_pLPU83a_0137. Strikingly, the values for the anti-activator *traM* were not significantly affected, while *rcgA* was down-regulated in the absence of RcgR.

Preliminary studies did not show changes in *traR* expression in the *rcgA* mutant (28). Nevertheless, the Dtr and Mpf gene expression could change when *rcgA* is mutated, leading to a non-conjugative phenotype. Unexpectedly, the RNAseq approach did not show any differentially expressed genes related to CT in the *rcgA* mutant, showing that the essential function of RcgA is not related to the transcription of conjugative genes.

## RcgR function is not related to anti-activator TraM

The *traM* gene encodes a protein usually involved in CT repression. In some described systems, TraM binds to TraR, generating an allosteric conformational change that prevents the binding of TraR to DNA targets (21, 44). Therefore, when TraM is expressed, the CT process is repressed. However, when TraR is not interacting with TraM, it is able to participate in a positive feedback loop that increases its own expression (45). Therefore, the interaction with TraM also regulates TraR levels.

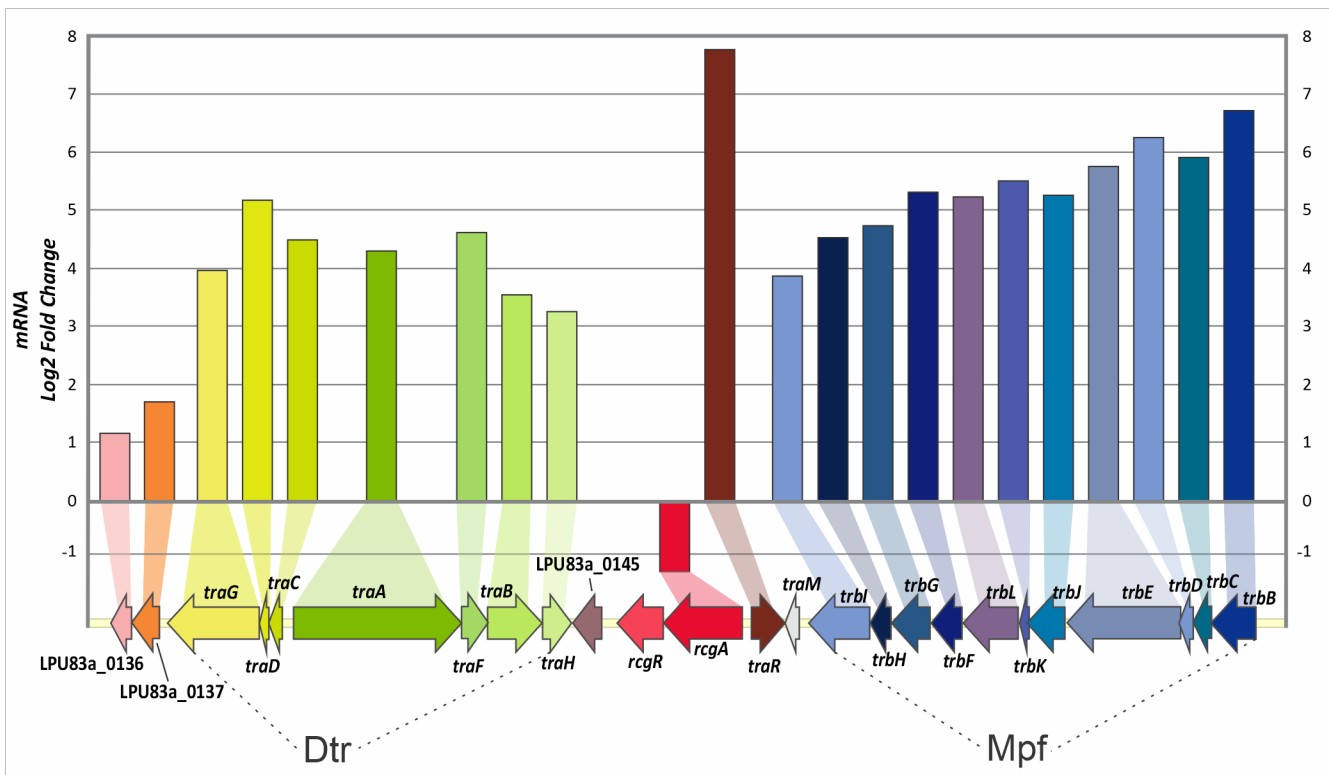

**FIG 2** Organization of conjugative genes of pLPU83a and fold change of the genes in the *rcgR* mutant. The genes affected by the deletion of *rcgR* and their organization are shown. *Y* axis shows the Log₂ Fold Change of each gene. The Dtr genes are shown in green ranges, and those belonging to Mpf are shown in blue ranges. The *P*-adjusted value of each gene is shown in Table 1.

The RNAseq experiment confirmed that a decrease in the expression of *traR* and Mpf and Dtr genes occurs when RcgR is active, a behavior that could be similar to the effect of the anti-activator TraM. Despite *traM* being expressed in wild type and the *rcgR* mutant (see Table S5 at https://doi.org/10.35537/10915/189658, raw counts), its expression is not affected in the *rcgR* mutant. Thus, we hypothesized that the repression generated by RcgR could be similar to the effect of TraM, either by interacting with TraM or as a second independent negative regulator. To study this hypothesis, pLPU83aΔ*traM* and pLPU83aΔ*traM*Δ*rcgR* strains were generated. As shown in Fig. 3, deletion of *traM* did not affect pLPU83a conjugative transfer frequencies, neither in the wild-type strain nor in the *rcgR* mutant, demonstrating that RcgR function does not rely on TraM, but also that TraM is not functional under these conditions.

## Activation by TraR cannot be achieved by deletion of *rcgR*

TraR is essential for pLPU83a conjugative transfer in its native background (31). Due to Δ*rcgR* showing increased expression of *traR* and the Dtr and Mpf genes, we evaluated whether the plasmid with a deletion in *rcgR* was able to be transferred even in the absence of TraR. This could be possible considering a direct activation of Dtr and Mpf genes due to the lack of *rcgR*. To analyze this possibility, a double mutant pLPU83aΔ*traR*Δ*rcgR* was constructed and tested regarding its CT behavior. While pLPU83aΔ*rcgR* transfers at a CT frequency of $4.93 \pm 1.63 \times 10^{-6}$ transconjugants/donor cell, pLPU83aΔ*traR*Δ*rcgR* was not able to be transferred ($<5 \times 10^{-10}$ transconjugants/donor cell), confirming that TraR is completely necessary for the conjugation of pLPU83a and that the repression by RcgR is hierarchically below TraR.

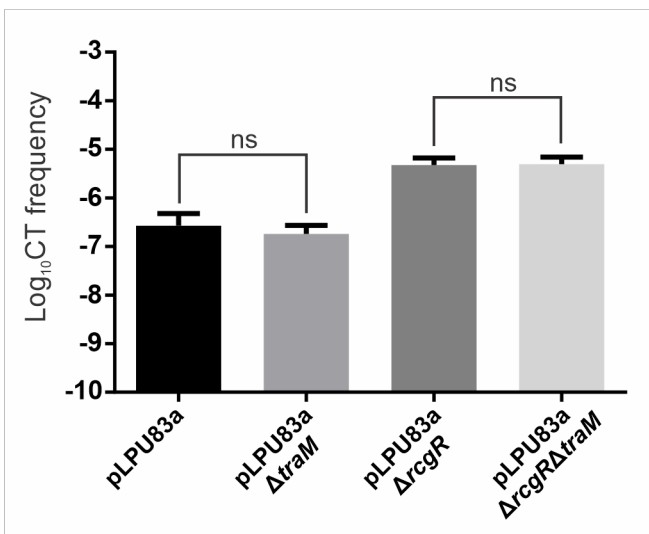

**FIG 3** Effect of *traM* deletion on pLPU83a conjugative frequencies. The conjugative frequencies obtained for the pLPU83aΔ*traM* mutant strain and the pLPU83aΔ*rcgR*Δ*traM* double mutant are shown. Statistical analysis using ANOVA did not show significant differences (ns) between the wild-type strain and the *traM* mutant, neither between the *rcgR* mutant strain and the *rcgR-traM* double mutant.

## DISCUSSION

Bacteria's adaptation to changing environmental conditions has been mainly driven by the acquisition of traits by HGT (46). One of the main mechanisms involved in HGT is plasmid conjugation (47). Although many efforts and much research have been done since its discovery, new actors are continuously described, either as part of the regulation or the mechanism itself. Advances in sequencing technologies have led to massive amounts of genomic data with a large proportion of genes annotated as hypothetical, whose roles remain unknown. The search for patterns and domains led us to approach the function of many proteins, although there are other ones with no associated function (i.e., hypothetical proteins). Moreover, in some cases, the function of some proteins may not be exactly as expected. For example, proteins of the LuxR family, such as TraR in rhizobial plasmid conjugation, have been widely described as a LuxR-type transcriptional regulator that responds to AHLs, regulating plasmid transfer based on a QS mechanism (TraR from *Agrobacterium tumefaciens*, AAA64793 or TraR from *Rhizobium etli*, AAO43545.1). Nevertheless, pLPU83a from *R. favelukesii* encodes a *traR* gene, similar to those present in QS-regulated plasmids, which does not respond to AHLs (the identity between proteins is 40.17% with TraR from *A. tumefaciens* and 24.69% with TraR from *R. etli*) (30). For these reasons, the characterization of proteins by biochemical and molecular biology approaches is relevant. Then, the obtained data could be extrapolated to similar systems, allowing a global understanding of them. pLPU83a and the other plasmids present in rhizobial group I-C show some differences from other rhizobial plasmids (30), which allow us to think that we are facing a new molecular regulatory system. In this work, we extend the characterization of RcgA and RcgR, two proteins involved in conjugative transfer and conserved in rhizobial plasmids.

In some cases, the lack of genes could affect a specific process without them being directly linked to it. Guynet et al. (37) described two proteins, StbA and StbB, encoded in an operon and found that their function is related to the stability of the plasmid in which they are encoded. Nevertheless, deletion of these genes generates opposite effects on plasmid transfer behavior. In this study, we showed that, although StbA/StbB and RcgR/RcgA share a similar genetic organization in operons and their deletions both impact plasmid transfer, the RcgR/RcgA system does not affect the stability of pLPU83a.

Western blot analysis showed that RcgR was present in both the cytosolic and membrane fractions, suggesting dual localization. Consistently, fluorescence microscopy revealed variable localization patterns, with the protein appearing either in the cytoplasm or at the membrane despite the absence of a transmembrane domain, potentially indicating transient membrane association or dynamic relocalization. In the Dot/Icm type 4B secretion system of *Legionella pneumophila*, it has been described that DotO (a distant homolog of VirB4 ATPase) associates with the bacterial inner membrane through three Dot/Icm inner membrane components (48), so an interaction with Mpf components could explain this localization. Alternatively, another explanation for this dual localization could be a function related to the connection between the relaxosome and the membrane, similar to that described for the partition protein ParB in the *Neisseria gonorrhoeae* T4SS (49).

Regarding the expression of conjugation-related proteins, we confirmed that TraR and several proteins of Dtr and Mpf were more abundant in the absence of RcgR than in the wild-type strain. This correlates with the transcriptional data, where *traR* was the gene with the highest fold change in the Δ*rcgR* mutant, followed by all the conjugative genes. The proteomic and transcriptomic results could suggest two main pathways in the regulation mediated by RcgR. On the one hand, RcgR could act as a transcriptional repressor, directly inhibiting the expression of genes related to CT (either by directly inhibiting transcription of *traR*, *tra,* and *trb* genes, or inhibiting TraR and, as a consequence, *tra* and *trb* genes) (2 in Fig. 4). On the other hand, RcgR could inhibit the function of TraR, maintaining the transcription of conjugative genes at a basal level (3 in Fig. 4). Then, when RcgR is absent, TraR is active and leads to an increase in the expression of Dtr and Mpf genes (45, 50). Both mechanisms would be supported by the fact that the

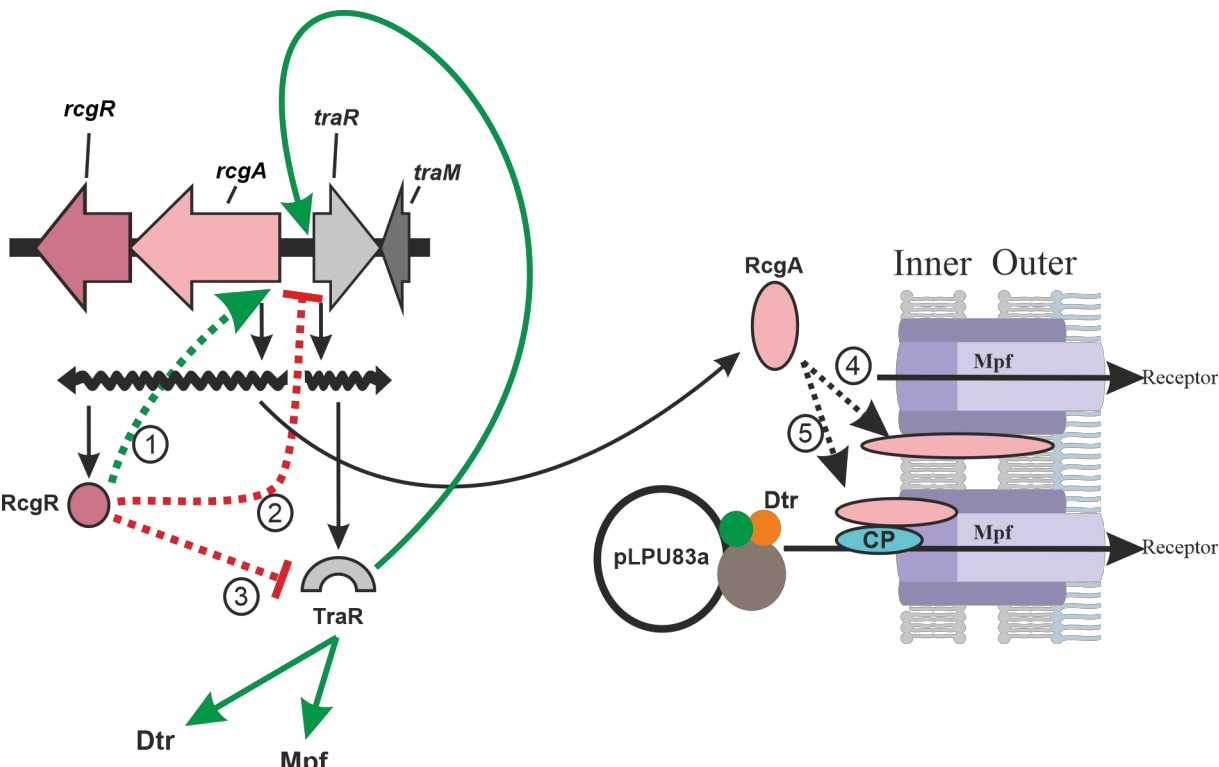

**FIG 4** Hypothetical models of RcgR and RcgA functioning. Proposed models for RcgR (1). RcgR binds to its own promoter, activating its own transcription (2). As a consequence of transcriptional interference generated by RcgR binding, *traR* transcription is affected (3). RcgR and TraR could interact, leading to a decreased activity of TraR. Proposed models for RcgA (4). RcgA could act as a structural protein necessary for Mpf assembly (5). RcgA could be involved in the recruitment of the relaxosome to the T4SS. Continuous green arrows indicate upregulation of the genes described in bibliography. Dotted lines show possible functions for RcgA and RcgR (green: activation, red: inhibition, and black: localization). CP, coupling protein.

transcription of *traR* presents the highest fold change when RcgR is absent, which would indicate that *traR* levels respond to the absence of RcgR. Nevertheless, these results do not explain whether Mpf and Dtr genes respond directly to the lack of RcgR or the rise of TraR levels. The non-conjugative phenotype of the mutant pLPU83aΔ*rcgR*Δ*traR* confirmed that the overexpression of Dtr and Mpf genes should be mediated by TraR (Fig. 4) and not by direct inhibition of RcgR over Dtr and Mpf genes. Nevertheless, a weak interaction between RcgR and the promoters of Dtr and Mpf genes could not be disregarded. Binding assays of RcgR to these promoters could answer this question.

Despite RcgA being totally needed for CT, its transcription was downregulated in the absence of RcgR, even when the CT frequency of the *rcgR* mutant is higher than that of the wild type. Considering that *rcgAR* are in an operon, RcgR could regulate its own expression in a positive feedback loop (1 in Fig. 4), in a similar way as described for TraR in other systems (45, 50). This would suggest that RcgR, despite the lack of a known DNA-binding domain, might bind to DNA. Given that the *rcgAR* operon and *traR* are divergently transcribed and they share the intergenic region (where the promoters are located), a consequence of this positive feedback loop could be that RcgR interferes with *traR* transcription (2 in Fig. 4). Similarly, this happens in another rhizobial plasmid conjugative system, where *rctA* is located divergently to the *virB* operon (which encodes for the Mpf system in the symbiotic plasmid of *R. etli* CFN42), and the expression of RctA transcriptionally represses the expression of the *virB* operon (16). Notably, analogous regulatory strategies have been described in gram-positive systems, such as the pIP501 plasmid of *Enterococcus faecalis*, where the small cytosolic protein TraN (encoded within the *tra* conjugative transfer operon) represses conjugative transfer by binding to a specific promoter of the *tra* operon and downregulating the expression of conjugative genes (51). These parallels suggest that direct transcriptional repression by plasmid-encoded regulators may represent a conserved mechanism across diverse conjugative systems. Alternatively, the observed repression of *rcgAR* in the *rcgR* mutant could result from increased TraR expression, which in turn might repress *rcgAR* by binding to the shared promoter region. Further experiments are needed to distinguish between these two non-exclusive scenarios. The RNAseq data also allowed us to observe that the anti-activator *traM* gene is transcribed (LPU83_pLPU83a_0150, see Table S5 at https://doi.org/10.35537/10915/189658), although transcript levels are not showing significant differences between the wild type and the *rcgR* mutant. In addition to this, derivatives of pLPU83a-13 and pLPU83aΔ*rcgR* with *traM* deletions showed similar conjugative frequencies to the parental strains, confirming that TraM would not be involved in the regulation of the process, at least under these conditions, similarly to the behavior described in *R. etli* (19).

The transcriptomic analysis revealed other differentially expressed genes in the *rcgR* mutant. Three genes encoded in pLPU83a were overexpressed in the mutant strain. In particular, two of them caught our attention since they are encoded in an operon next to the Dtr of the plasmid; hence, they may be involved in the process. One of them, LPU83_pLPU83a_0136, encodes a putative thermonuclease family protein, while LPU83_pLPU83a_0137 encodes a hypothetical protein with a WGR domain (a protein domain named after its conserved tryptophan–glycine–arginine motif), frequently associated with a DNA-binding domain. Future studies of these genes must be performed to clarify their function.

As we previously reported, RcgA is essential for pLPU83a transfer. Data obtained in this work suggest that its function is not related to the regulation of the actors involved in the process, since no conjugative genes were differentially expressed in either the proteomic or the transcriptomic analysis of pLPU83aΔ*rcgA*. Both Western blot and microscopy experiments confirmed a membrane localization of RcgA, consistent with previously predicted transmembrane domains (28). Specifically, a continuous localization through the membrane was observed (Fig. 1G through I). A comparable pattern has been reported for the coupling protein of plasmid R388 (52). Assuming that the CP of pLPU83a has a similar distribution pattern and the fact that both proteins are

essential for conjugation, a related function or even a direct interaction between RcgA and the CP of pLPU83a is plausible (5 in Fig. 4). In addition, proteins of the *vir*-T4SS of pTI plasmid from *A. tumefaciens* have been localized to multiple foci in a helical or periodic pattern around the circumference of the cells (53). Since the deletion of *rcgA* did not alter the transcription of conjugative genes or protein abundance, a structural role in Mpf formation remains possible (4 in Fig. 4). Although the core components of type IV secretion systems in gram-negative bacteria have been extensively characterized, additional proteins may contribute to the assembly, stabilization, or regulation of these complexes. Supporting this idea, the small membrane protein TraB, encoded by plasmid pIP501 of the gram-positive bacterium *E. faecalis*, was shown to be essential for conjugative transfer and to interact with multiple mating pair formation proteins, potentially forming part of the complex itself or acting as a recruitment factor (54).

Our results show that RcgR represses the expression of *traR*, thereby reducing CT. Future efforts will focus on elucidating the molecular mechanism underlying this repression. RcgA was previously established as essential for CT, and even though our current findings point to a membrane localization, its molecular function remains to be clarified. These findings mark a significant step forward in our understanding of the regulatory network that controls plasmid dissemination in soil bacteria, and further studies are needed to achieve a comprehensive understanding of these mechanisms.

## MATERIALS AND METHODS

### Bacterial strains and plasmids

The strains and plasmids used in this work are listed in Table S1 (https://doi.org/10.35537/10915/189658). *Escherichia coli* was grown in LB medium (55) at 37°C. *Rhizobium* and *Agrobacterium* strains were grown on TY (56) at 28°C. For solid media, 15 g of agar per liter of medium was added. The final concentration of antibiotics used was 10 µg mL$^{-1}$ gentamicin and 25 µg mL$^{-1}$ kanamycin for *E. coli*. For *Rhizobium* and *Agrobacterium,* 400 µg mL$^{-1}$ streptomycin, 120 µg mL$^{-1}$ neomycin (Nm), 200 µg mL$^{-1}$ rifampicin, 2.5 µg mL$^{-1}$ tetracycline, and 30 µg mL$^{-1}$ gentamicin.

### Bacterial matings

Bacterial matings were performed as described by Simon et al. (57). Briefly, overnight cultures were grown to the stationary phase. Donor and recipient strains were mixed at a 1:1 ratio, plated onto TY plates, and incubated overnight at 28°C. Bacteria were resuspended in 1 mL of 10 mM MgSO$_4$-0.01% Tween 40 (vol/vol). Serial dilutions were plated on selective TY agar supplemented with the corresponding antibiotics to quantify the number of donor, recipient, and transconjugant cells. The conjugation frequencies are calculated as the ratio of transconjugants per donor cell. In each case, plasmid profiles to corroborate the desired plasmid transfer were done by Eckhardt gels (58) as modified by Hynes and McGregor (59).

### DNA manipulation, genetic constructs, and mutagenesis

Total DNA and plasmid preparations, restriction-enzyme analysis, cloning procedures, and *E. coli* transformation were performed according to previously established techniques (60).

PCR amplification was carried out with recombinant *Taq* DNA polymerase or *Phusion* DNA polymerase as specified by the manufacturers. The primers used in this study are listed in Table S2 (https://doi.org/10.35537/10915/189658).

#### *Plasmid constructions for mutagenesis and generation of deletional mutants*

For the construction of the vector to generate a deletion in *traM* from pLPU83a, two fragments of *traM* were amplified with *Phusion* polymerase and primers *traM-Nter-Bam/*

*traM-Nter-Xba* (218 bp) and *traM-Cter-Bam/traR-left-Xho* (260 bp). Both fragments were cloned into the *SmaI* site of pK18mob, obtaining pK18::*traM*Nter and pK18::*traM*Cter, respectively. pK18::*traM*Nter was digested with *XbaI* and *BamHI,* and the released fragment was cloned into the *XbaI/BamHI* sites of pJQ200KS, obtaining pJQ200::*traM*Nter. Then, pK18::*traM*Cter was digested with *BamHI* and *XhoI,* and the released fragment was cloned into pJQ200::*traM*Nter, obtaining pJQ200::*traM* (5,794 bp).

For the construction of the vector to generate a deletion in *traR*, a fragment of *traR* was amplified with *Phusion* polymerase and primers *traR-Fw-out/traR-Rv-out* (876 bp). The fragment was cloned into the *SmaI* site of pK18mob, obtaining pK18::*traR*. pK18::*traR* was digested with *EcoRV* and re-ligated in a larger volume, removing an internal fragment of 439 bp and obtaining pK18::Δ*traR*. pK18::Δ*traR* was digested with *SacI* and *BamHI,* and the released fragment was cloned into the *SacI/BamHI* sites of pJQ200KS, obtaining pJQ200::Δ*traR* (5,783 bp).

The constructed vectors were introduced by conjugation into *R. favelukesii* LPU83-13 or *R. favelukesii* LPU83-13Δ*rcgR*. Double recombinants were selected in TY-sucrose 10% (wt/vol) medium as neomycin resistant and gentamicin sensitive.

### Plasmid constructions for GFP fusions and generation of complemented strains

Fusions between RcgA or RcgR and GFP protein were achieved by a two-step PCR. For the design of the primers, we employed a 24 bp hybridization sequence, added both in the C-terminal region of *rcgA* or *rcgR* and in the N-terminal region of GFP. This sequence encodes a flexible linker composed of two repeats of Gly-Gly-Gly-Ser. In the first step, each gene was amplified with *Phusion* polymerase and primers *rcgA-Nter-Kpn/rcgA-Cter-GFP* (*rcgA,* 1,864 bp), *rcgR-Nter-Kpn/rcgR-Cter-GFP* (*rcgR,* 1,111 bp), and *GFP-Nter/GFP-Cter-Xba* (GFP, 756 bp). In the second step, equimolar amounts of *rcgA* or *rcgR* and GFP were mixed with *Phusion* polymerase in the absence of primers, allowing hybridization of complementary regions and extension of the fragments. Then, corresponding primers were added, allowing the amplification of the full fragments (*rcgA*::GFP, 2,594 bp; *rcgR*::GFP, 1,842 bp). The obtained fragments were cloned into the *SmaI* site of pBBR1MCS-5, obtaining pBBR1MCS-5::*rcgA*::GFP (7,268 bp) and pBBR1MCS-5::*rcgR*::GFP (6,547 bp). As a control, GFP was amplified with *Phusion* polymerase and primers *GFP-Nter/GFP-Cter-Xba*, and the obtained fragment was cloned into the *SmaI* site of pBBR1MCS-5, obtaining pBBR1MCS-5::GFP (5,524 bp).

To check the functionality of the constructions, the constructed vectors were transformed into *E. coli* S17-1, followed by biparental matings to the respective mutant strains. Transconjugants were selected as neomycin and gentamicin resistant.

### Stability assays

Each strain was inoculated in VMM (61) liquid media without antibiotics. Once the cultures reached the stationary phase, each strain was re-inoculated in fresh media, repeating the process four times. Then, after approximately 40 generations, serial dilutions of each culture were plated on TY solid media, with and without Nm. The stability is expressed as the quotient between the number of colonies containing the plasmid (grown in TY-Nm) and the total number of colonies (grown in TY without antibiotics).

### Microscopy

For image acquisition, samples of each strain were collected after overnight growth on TY media. All samples were washed with a solution of $MgSO_4$ (10 mM) and Tween 40 (0.001% [vol/vol]) and resuspended in PBS (1×). Samples were incubated for 5 min with 10 µM of FM4-64 (Invitrogen) followed by an additional wash with $MgSO_4$–Tween 40 solution. The cells were immediately mounted onto a microscope using a coverslip (#1.5 Thickness, ThorLabs).

Fluorescence images were acquired using the Nanoimager S microscope (Oxford Nanoimaging) with a 100× objective (NA = 1.4, oil immersion). A total of 100 frames

were acquired per field of view using excitation lasers 473 nm (15% laser power) and 561 nm (15% laser power), with an exposure time of 30 ms per frame. Three independent biological samples were acquired per strain.

The resolution of all images was enhanced using the SRRF algorithm (36) implemented in FIJI, with the following parameters: ring radius = 0.5, radial magnification = 10, and axes in ring = 8.

Intensity profiles were generated in FIJI from the super-resolved images by defining a line through the cell cytoplasm, with the boundaries marked at each cell pole. Due to variations in cell length, measurements were normalized between −1 and 1, with 0 representing the midpoint of each cell.

## Protein extraction for Western blotting

Cultures were grown on TY media until they reached an $OD_{600}$ of 1.5. A volume of 250 mL of each culture was centrifuged at 11,000 × $g$ for 30 min at 4°C and washed twice with low-salt washing buffer (3 mM KCl, 1.5 mM $KH_2PO_4$, 68 mM NaCl, and 9 mM $NaH_2PO_4$) (62). The pellets were resuspended in 6.5 mL of 10 mM Tris buffer, pH 7.6, with the protease inhibitor phenylmethylsulfonyl fluoride. The samples were transferred to 2 mL tubes with glass beads (a mixture of diameters 0.1–0.3 mm) and disrupted with a Precellys tissue homogenizer (Bertin Instruments), performing two cycles of 5,000 rpm for 20 s and cooling the samples on ice after each cycle. The samples were then centrifuged at 10,000 × $g$ for 20 min at 4°C, and the supernatants were transferred to new tubes for a subsequent 1-h incubation with DNase (RQ1 DNase, Promega) and RNase A at 37°C. The samples were next ultracentrifuged at 100,000 × $g$ for 2 h at 4°C. For each cell culture, two kinds of protein samples were obtained: a fraction enriched in membrane proteins and another in cytosolic proteins. The pellets containing the membrane proteins were resuspended in rehydration buffer (7 M urea, 2 M thiourea, 10% [vol/vol] isopropyl alcohol, and 2% [vol/vol] Triton X-100), whereas the cytosolic proteins were precipitated with four volumes of cold acetone overnight at −20 °C and stored until processed. Finally, those cytosolic proteins were centrifuged at 10,000 × $g$ at 4°C for 20 min, followed by a washing with 90% (vol/vol) acetone and resuspended in rehydration buffer (7 M urea and 2 M thiourea). The total protein content of the samples was determined by the Bradford assay with Coomassie Brilliant Blue (63).

## Western blotting

Samples were subjected to reducing SDS-PAGE and transferred onto polyvinylidene difluoride membranes by Western blotting, using a Mini *Trans*-Blot transfer Cell (BioRad). Proteins were detected on the membrane using GFP antibody (3H9) (Chromotek). Membrane-bound antibodies were detected using Rat IgG HRP-conjugated Antibody (R&D Systems). Images were acquired on a transilluminator with a digital camera HCCD BioChemi System Video (UVP Inc., USA).

## Shotgun proteomics analyses

Protein isolation and trypsin digestion were performed as described by Castellani et al. (28).

LC-MS/MS measurements were carried out using a Thermo QExactive Plus 3000 mass spectrometer (Thermo Fisher Scientific, Waltham, MA, USA) online coupled to the UltiMate LC system. The peptides were separated on a 25 cm steel column Acclaim PepMap 100 C18-LC-column with a particle size of 2 µm and a diameter of 75 µm (Thermo Fisher, MA, USA). Identification was performed using the software DIA-NN 1.8 with default settings, and the output was filtered at 0.01 FDR (64).

The statistical analysis of DIA-NN data was performed with Perseus 2.0.11 (65). A total of 3,727 were identified. Among them, in 64, the normalized intensities became null in all the samples. Thus, the final protein list contains 3,663 proteins.

For the extraction of extracellular proteins, 50 mL of cell cultures was centrifuged at 13,000 rpm for 5 min, and supernatants were filtered with 0.2 µm filters and frozen

in $N_2$. Then, the method developed by Wendler et al. (43) was followed. Briefly, the proteins from the freeze-dried supernatant were isolated by phenol extraction, followed by subsequent methanol precipitation and several washing steps with 70% ethanol. After drying, the extracellular proteins were resuspended in 200 µL TE-Buffer and 5 mM dithiothreitol (DTT) and incubated for 60 min at 60°C. Twenty microliters of 200 mM iodacetamide was added and left for 90 min in the dark. Next, 5 µL of 200 mM DTT was added to the solution and incubated for an additional 60 min. Extracellular proteins were digested following the same protocol as for the cellular fraction. The statistical analysis of DIA-NN data was performed with Perseus 2.0.11. A total of 3,727 proteins were identified. Among them, in 2,591, the normalized intensities became null in all the samples. Thus, the final protein list contains 1,136 proteins.

## RNA extraction

For RNA isolation, three biological replicates of *R. favelukesii* LPU83-13, *R. favelukesii* LPU83-13Δ*rcgA,* and *R. favelukesii* LPU83-13Δ*rcgR* were cultivated in TY media. Then, RNAprotect reagent (Qiagen, Hilden, Germany) was added, cells were harvested, and the pellets were frozen in liquid nitrogen. The commercial RNeasy R Protect Bacteria Mini Kit (Qiagen, Hilden, Germany) was used for the RNA extraction, and DNA was removed with DNase I (Qiagen, Hilden, Germany). Ribosomal RNA depletion was performed with a commercial kit (Ribo-Zero rRNA Removal Kit, Illumina Inc., San Diego, USA).

## RNA sequencing and data analysis

The cDNA library was made with the commercial kit TruSeqmRNA Sample Preparation (stranded) (Illumina Inc., San Diego, CA, USA). The samples were sequenced by means of Illumina HiSeq platforms available at the Center for Biotechnology (CeBiTec, Bielefeld University, Germany). After that, the quality of the sequences was evaluated with FastQC (http://www.bioinformatics.babraham.ac.uk/projects/fastqc/), and the reads were mapped unambiguously to the reference genome of *R. favelukesii* LPU83 (66) with Bowtie2 (67). Finally, the results were visualized through ReadXplorer 2.0 (68). The differential expression analysis was performed in R version 4.4.3 (R Core Team [2025]. R: A language and environment for statistical computing. R Foundation for Statistical Computing, Vienna, Austria. Available at https://www.R-project.org/) using DESeq2 1.46.0 (69), employing the function contrast (*P*-value of 0.05 and LFCthreshold of 0.585) and then LFCshrinkage type "ashr" (70). After this analysis, those genes with a $Log_2$ Fold Change >1 or <−1 and an adjusted *P*-value < 0.05 were considered differentially expressed between the conditions. Three differentially expressed genes were found in both mutants and in another mutant not shown in this work, with similar $Log_2$ Fold Change values; therefore, they were not considered in this analysis. In the LPU83-13Δ*rcgA* strain, ribosomal genes found in the transcriptome were not considered. The resulting raw count files for the nine transcriptomes are shown in Table S5 (https://doi.org/10.35537/10915/189658).

## ACKNOWLEDGMENTS

This work was partially supported by grants PICT2020-02314, PICT2021-00153, and PIP0678 to G.T.T. and PICT2020-02529 to J.P.-G. L.G.C. is an MSCA-Cofund fellow. M.D.C. and A.L. are fellows of CONICET. J.P.-G., M.P., and G.T.T. are members of the Research Career of CONICET. G.T.T. also acknowledges Alexander von Humboldt Foundation and Fundación Williams (Project No. 1045).

We also extend our heartfelt thanks and special acknowledgment to Dr. Susana Brom for her invaluable support and guidance.

## AUTHOR AFFILIATIONS

[1]Departamento de Ciencias Biológicas, Facultad de Ciencias Exactas, Instituto de Biotecnología y Biología Molecular (CCT-La Plata-CONICET), Universidad Nacional de La Plata, La Plata, Argentina

[2]Department of Biology, University of Pennsylvania, Philadelphia, Pennsylvania, USA

[3]Departamento de Microbiología Molecular, Instituto de Biotecnología, Universidad Nacional Autónoma de México, Cuernavaca, Mexico

[4]Genome Research of Industrial Microorganisms, Center for Biotechnology (CeBiTec), Bielefeld University, Bielefeld, Germany

## PRESENT ADDRESS

Lucas G. Castellani, Institute for Multidisciplinary Research in Applied Biology (IMAB), Universidad Pública de Navarra (UPNA), Pamplona, Spain

## AUTHOR ORCIDs

Lucas G. Castellani http://orcid.org/0000-0002-4678-7385
María Delfina Cabrera http://orcid.org/0009-0007-7320-8864
Juliet F. Nilsson http://orcid.org/0000-0003-4203-5263
Julieta Pérez-Giménez http://orcid.org/0000-0002-5979-9596
Tobias Busche https://orcid.org/0000-0001-9211-8927
Andreas Schlüter https://orcid.org/0000-0003-4830-310X
Karsten Niehaus https://orcid.org/0000-0003-4078-9870
Mariano Pistorio https://orcid.org/0000-0001-7931-8721
Gonzalo Torres Tejerizo http://orcid.org/0000-0002-2508-0292

## FUNDING

| Funder | Grant(s) | Author(s) |
| --- | --- | --- |
| Consejo Nacional de Investigaciones Científicas y Técnicas | PIP2020-0678 | Gonzalo Torres Tejerizo |
| Agencia Nacional de Promoción Científica y Tecnológica | PICT2021-0153, PICT2020-02314 | Gonzalo Torres Tejerizo |
| Agencia Nacional de Promoción Científica y Tecnológica | PICT2020-02529 | Julieta Pérez-Giménez |
| Fundacion Williams | Project Nº 1045 | Gonzalo Torres Tejerizo |

## AUTHOR CONTRIBUTIONS

Lucas G. Castellani, Conceptualization, Data curation, Formal analysis, Investigation, Methodology, Writing – original draft, Writing – review and editing | María Delfina Cabrera, Conceptualization, Data curation, Formal analysis, Investigation, Methodology, Writing – original draft, Writing – review and editing | Abril Luchetti, Conceptualization, Data curation, Formal analysis, Writing – review and editing | Juliet F. Nilsson, Conceptualization, Data curation, Formal analysis, Investigation, Writing – review and editing | Julieta Pérez-Giménez, Conceptualization, Data curation, Formal analysis, Investigation, Methodology, Writing – review and editing | Luis Alfredo Bañuelos-Vazquez, Investigation, Methodology, Writing – review and editing | Alma Alva, Formal analysis, Methodology, Writing – review and editing | Daniel Wibberg, Conceptualization, Data curation, Formal analysis, Investigation, Writing – review and editing | Tobias Busche, Conceptualization, Data curation, Formal analysis, Methodology, Writing – review and editing | Jörn Kalinowski, Investigation, Resources, Supervision, Writing – review and editing | Andreas Schlüter, Conceptualization, Investigation, Methodology, Supervision, Writing – review and editing | Alfred Pühler, Conceptualization, Funding acquisition, Investigation,

Supervision, Writing – review and editing | Karsten Niehaus, Conceptualization, Formal analysis, Funding acquisition, Investigation, Writing – review and editing | Mariano Pistorio, Conceptualization, Methodology, Supervision, Writing – review and editing | Gonzalo Torres Tejerizo, Conceptualization, Data curation, Formal analysis, Funding acquisition, Investigation, Methodology, Project administration, Supervision, Validation, Writing – original draft, Writing – review and editing

## DATA AVAILABILITY

The mass spectrometry proteomics data have been deposited to the ProteomeXchange Consortium via the PRIDE (71) partner repository with the data set identifier PXD067087. Raw transcriptome data have been deposited under SRA data accession PRJNA1301829.

## ADDITIONAL FILES

The following material is available online.

### Open Peer Review

**PEER REVIEW HISTORY (review-history.pdf).** An accounting of the reviewer comments and feedback.

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
