## [Reviewer comments · Microbiology Spectrum]

Microbiology Spectrum

Characterization of RcgA and RcgR, two rhizobial proteins involved in the modulation of plasmid transfer

Lucas Castellani, Maria Cabrera, Abril Luchetti, Juliet Nilsson, Julieta Pérez-Giménez, Luis Alfredo Bañuelos-Vazquez, Alma Alva, Daniel Wibberg, Tobias Busche, Jörn Kalinowski, Andreas Schlüter, Alfred Pühler, Karsten Niehaus, Mariano Pistorio, and Gonzalo Torres Tejerizo

Corresponding Author(s): Gonzalo Torres Tejerizo, Instituto de Biotecnología y Biología Molecular

Review Timeline:

Submission Date:	October 10, 2025
Editorial Decision:	November 20, 2025
Revision Received:	November 27, 2025
Accepted:	December 24, 2025

Editor: Nicolas Soler

Reviewer(s): Disclosure of reviewer identity is with reference to reviewer comments included in decision letter(s). The following individuals involved in review of your submission have agreed to reveal their identity: Elisabeth Grohmann (Reviewer #1)

Transaction Report:

DOI: <https://doi.org/10.1128/spectrum.03242-25>

Re: Spectrum03242-25 (**Characterization of RcgA and RcgR, two rhizobial proteins involved in the modulation of plasmid transfer**)

Dear Dr. Gonzalo Arturo Torres Tejerizo:

Thank you for the privilege of reviewing your work. Below you will find my comments, instructions from the Spectrum editorial office, and the reviewer comments.

Your work has now been evaluated by two experts in the field. Both reviewers acknowledged a significant contribution in understanding the regulation of bacterial conjugation. Importantly, reviewer 2 raised concerns about the description of some experiments in your results, especially about Figure 1 description and conclusions. In order to consider publishing your work, I ask that you respond to each point for correction suggested by the two experts.

Revision Guidelines

Sincerely,
Nicolas Soler
Editor
Microbiology Spectrum

Reviewer #1 (Comments for the Author):

The manuscript of Castellani et al. is an interesting contribution on regulatory aspects of conjugative plasmid transfer in important soil bacteria. The authors propose the proteins RcgA and RcgR as key elements in rhizobial plasmid transfer regulation. In summary, based upon their data they propose a mechanism of TraR (the transcriptional regulator of conjugative transfer) control, which is uncoupled from Quorum Sensing-based signaling. The work advances our mechanistic understanding of plasmid transfer control in Rhizobia.

The study is a sound piece of work in which proper state of the art methodology has been applied to unravel the studied mechanisms. Proper controls and statistical methods have been applied. The manuscript is generally well written.

Minor comments:

l. 207, 210, 215, 217: "Column" instead of screen

l. 325: conjugation related proteins

l. 327: with the observation at the....instead of with "the observed".

l. 378: WGR domain. Define the abbreviation, please

l. 389-390:..... or even a direct

interaction between them can be hypothesized". What do you mean by this? Interaction of the CP of R388 and RcgA of a Rhizobium plasmid? Please clarify

l. 414: grown in

l. 423:... and incubated overnight at 28 degrees.

l. 425: TY agar

Throughout the whole manuscript: Only the first three characters of restriction enzymes should be italicized.

The Supplementary Material is an important addition to the main text.

One minor comment: In Table S1, not for all constructs the origin is given.

Reviewer #2 (Comments for the Author):

Catellani et al has here characterised RagA and RcgR, where RcgR inhibits traR expression and subsequently decreasing the expression of T4SS genes. RcgA is shown to be essential for transfer, and is suggested to instead play a structure role in the mating pair formation. Overall, the paper provides important new insights into the control of conjugation in these systems, but there are a couple points that this reviewer finds difficult to understand with the data and the connected text, which in my mind would need to be clarified.

Major points

* lines 162-163. The authors state that RcgA-GFP runs at >180 kDa on the Western Blot, despite being only 90 kDa. This is surprising, but there is no reasonable explanation given. I can't find in the materials and methods whether this was a reducing SDS-PAGE, but I'm assume that was the case? If not, have the authors tried to compare a non-reducing with a reducing SDS-PAGE to see if the higher than expected MW can be explained by a cysteine-bridge?

* lines 1762-172 & Fig. 1 & Fig S2. The authors state that the control with GFP was found in lying scattered in the cytoplasm, but in the figure it is clear that the GFP is found in defined compartments in the cell, and not throughout the entire cytoplasm as expected. What is going on here? The figure does not seem to show what the authors state in the text, and it's also not the usual behaviour or free GFP in bacterial cells. The quantification of free GFP signal in Fig S2 does not really help to explain the behaviour, as also there it seems that the GFP is not randomly distributed throughout the cells.

Further, it seems that RcgR in the WB experiment is predominantly associated with the membrane fraction, but that does not quite seem to be the case in the fluorescence microscopy images in Fig 1D-F? Could the authors please explain what is going on here? Have the authors tried any standard techniques to check if the RcgR behaves as a typical peripheral membrane protein?

* Discussion, lines 381-401, including fig 4. While I do usually appreciate speculation in the discussion section, I find this part a bit too speculative. I think the right panel of Fig 4 and the related discussion text a bit too loose, as the authors do not present any data at all regarding how RcgA could potentially interact with the T4SS. Yes, it is possible that it's interacting with the T4CP, and yes it could also be an integral part of the T4SS, but it could also be involved in the initial assembly but not the final functional structure, and several other things. I would suggest that the authors tone down this piece of the discussion.

Minor points

* lines 102-106. There is quite a bit in here that is not greatly explained, like what is LuxR. Further, AHL is stated to be produced

by TraI. However, for many readers from the conjugation field TraI is the relaxase in e.g. the F system and pKM101. Would be good to clarify it here to avoid confusion

AUTHOR'S REPLY TO THE REVIEWERS COMMENTS.

Manuscript ID: Spectrum03242-25

Title: Characterization of RcgA and RcgR, two rhizobial proteins involved in the modulation of plasmid transfer

The reviewer comments are in black

The answers are in red

Reviewer #1 (Comments for the Author):

The manuscript of Castellani et al. is an interesting contribution on regulatory aspects of conjugative plasmid transfer in important soil bacteria. The authors propose the proteins RcgA and RcgR as key elements in rhizobial plasmid transfer regulation. In summary, based upon their data they propose a mechanism of TraR (the transcriptional regulator of conjugative transfer) control, which is uncoupled from Quorum Sensing-based signaling. The work advances our mechanistic understanding of plasmid transfer control in Rhizobia.

The study is a sound piece of work in which proper state of the art methodology has been applied to unravel the studied mechanisms. Proper controls and statistical methods have been applied. The manuscript is generally well written.

We thank the reviewer for the commentaries.

Minor comments:

l. 207, 210, 215, 217: "Column" instead of screen

We did not find the word "screen" in those lines, but we assume the reviewer is referring to the term "sheet." We would prefer to keep the word "sheet," as we are referring to different data sheets within the Excel file rather than to columns within the same data sheet.

l. 325: conjugation related proteins

Accepted. The correction was done.

l. 327: with the observation at the....instead of with "the observed".

Accepted. The correction was done.

l. 378: WGR domain. Define the abbreviation, please

We would like to clarify that "WGR domain" is not an abbreviation but the established name of a protein domain, derived from its conserved tryptophan-glycine-arginine (W-G-

R) motif. We have now added this explanation in the revised manuscript to improve its clarity.

Now, Lines 380-382: “LPU83_pLPU83a_0137 encodes a hypothetical protein with a WGR domain (a protein domain named after its conserved tryptophan–glycine–arginine motif), frequently associated with a DNA binding domain

l. 389-390:..... or even a direct interaction between them can be hypothesized". What do you mean by this? Interaction of the CP of R388 and RcgA of a Rhizobium plasmid? Please clarify

We referred to a possible interaction between RcgA and the CP of pLPU83a plasmid, considering that the distribution of Rhizobium CP is similar to that of R388. It was clarified on the main text as follows:

Now, Lines 391-394: “A comparable pattern has been reported for the coupling protein (CP) of plasmid R388 (52). Assuming that CP of pLPU83a have a similar distribution pattern and the fact that both proteins are essential for conjugation, a related function or even a direct interaction between RcgA and the CP of pLPU83a is plausible”

l. 414: grown in

Accepted.

l. 423:... and incubated overnight at 28 degrees.

Accepted.

l. 425: TY agar

Accepted.

Throughout the whole manuscript: Only the first three characters of restriction enzymes should be italicized.

We thank the reviewer for this observation. We carefully considered this point during manuscript preparation and followed the formatting commonly used in other articles published in the same journal (e.g. <https://journals.asm.org/doi/10.1128/spectrum.02941-24>, a recent published manuscript). To maintain consistency with the journal’s prevailing style, we would prefer to keep the current formatting, unless the editor advises otherwise.

The Supplementary Material is an important addition to the main text.

One minor comment: In Table S1, not for all constructs the origin is given.

We thank the reviewer for this observation. We checked and detected that the origin of pBBR1MCS-5::GFP construction was missed. It has been added on table S1.

Reviewer #2 (Comments for the Author):

Catellani et al has here characterised RcgA and RcgR, where RcgR inhibits traR expression and subsequently decreasing the expression of T4SS genes. RcgA is shown to be essential for transfer, and is suggested to instead play a structure role in the mating pair formation. Overall, the paper provides important new insights into the control of conjugation in these systems, but there are a couple points that this reviewer finds difficult to understand with the data and the connected text, which in my mind would need to be clarified.

We thank the reviewer for the commentaries. Many improvements have been made as suggested by the reviewer.

Major points

*** lines 162-163. The authors state that RcgA-GFP runs at >180 kDa on the Western Blot, despite being only 90 kDa. This is surprising, but there is no reasonable explanation given. I can't find in the materials and methods whether this was a reducing SDS-PAGE, but I'm assume that was the case? If not, have the authors tried to compare a non-reducing with a reducing SDS-PAGE to see if the higher than expected MW can be explained by a cysteine-bridge?**

We thank the reviewer for this comment, we also had concerns about the size as we mentioned in the text, but our goal was to find where the protein was located. The Western blot shown was indeed performed under reducing and denaturing conditions (SDS-PAGE with β -mercaptoethanol). We have now clarified this explicitly in the Materials and Methods section.

Regarding the apparent molecular weight of RcgA-GFP (>180 kDa when the predicted molecular weight is ~90 kDa), our original intention with this experiment was primarily to confirm the subcellular localization of the fusion protein. However, following the reviewer's question, we examined the literature more closely and found that anomalous migration of membrane proteins in SDS-PAGE has been widely reported. Several integral membrane proteins do not bind SDS in a uniform charge-to-mass ratio, leading to unexpected apparent molecular weights (A. Rath, M. Glibowicka, V.G. Nadeau, G. Chen, & C.M. Deber, Detergent binding explains anomalous SDS-PAGE migration of membrane proteins, Proc. Natl. Acad. Sci. U.S.A. 106 (6) 1760-1765, <https://doi.org/10.1073/pnas.0813167106> (2009) and A. Rath, F. Cunningham, & C.M. Deber, Acrylamide concentration determines the direction and magnitude of helical membrane protein gel shifts, Proc. Natl. Acad. Sci. U.S.A. 110 (39) 15668-15673, <https://doi.org/10.1073/pnas.1311305110> (2013)).

In addition, some proteins can form SDS-resistant oligomers, remaining dimeric during SDS-PAGE despite reducing and denaturing conditions. For example, Gentile et al. (Fabrizio Gentile, Pietro Amodeo, Ferdinando Febbraio, Francesco Picaro, Andrea Motta, Silvestro Formisano, Roberto Nucci, SDS-resistant Active and Thermostable Dimers Are Obtained from the Dissociation of Homotetrameric β -Glycosidase from Hyperthermophilic *Sulfolobus solfataricus* in SDS: STABILIZING ROLE OF THE A-C INTERMONOMERIC INTERFACE*, Journal of Biological Chemistry, Volume 277, Issue 46, 2002, Pages 44050-44060, ISSN 0021-9258, <https://doi.org/10.1074/jbc.M206761200>.) showed that a hyperthermophilic β -glycosidase forms a stable dimer that migrates at approximately twice the monomeric size in SDS-PAGE. Given that RcgA contains multiple predicted transmembrane helices, we cannot rule out the possibility that it forms a stable dimeric or partially oligomeric species that is not fully dissociated by SDS.

We have added this explanation to the revised manuscript.

Now Lines 163-167: “However, its apparent molecular weight was higher than anticipated (~90 kDa), migrating above the highest molecular weight marker (180 kDa). This higher-than-expected size may reflect the anomalous migration often reported for transmembrane proteins, which can bind SDS irregularly (33, 34), or the possibility that RcgA::GFP forms SDS-resistant dimers (35).

*** lines 1762-172 & Fig. 1 & Fig S2. The authors state that the control with GFP was found in lying scattered in the cytoplasm, but in the figure it is clear that the GFP is found in defined compartments in the cell, and not throughout the entire cytoplasm as expected. What is going on here? The figure does not seem to show what the authors state in the text, and it's also not the usual behaviour or free GFP in bacterial cells. The quantification of free GFP signal in Fig S2 does not really help to explain the behaviour, as also there it seems that the GFP is not randomly distributed throughout the cells.**

We thank the reviewer for the observation. On the one hand, the application of super-resolution techniques, deconvolution methods, and noise reduction algorithms effectively improves the signal-to-noise ratio. However, it is important to note that these methods can also amplify heterogeneities, thus transforming a diffuse signal into pronounced bright spots. Furthermore, the thickness of bacteria generally approaches the limits of optical resolution. Depending on factors such as position, focal plane, and membrane curvature, the intensity of cytosolic fluorescence may appear higher in specific regions or compartments, despite being uniformly distributed throughout the cell (See <https://journals.biologists.com/jcs/article/138/10/jcs263567/368116/Resolution-in-super-resolution-microscopy-facts> and <https://pubmed.ncbi.nlm.nih.gov/articles/PMC8808742/>). On the other hand, the purpose of showing different images together with the fluorescence-intensity quantification was to demonstrate that the signal is not restricted to a fixed polar spot. Instead, the fluorescence is distributed along the entire cell. Fig S2 shows that in different analyzed cells, GFP is randomly located in comparison with RcgA::GFP that is always in the membrane.

Further, it seems that RcgR in the WB experiment is predominantly associated with the membrane fraction, but that does not quite seem to be the case in the fluorescence microscopy images in Fig 1D-F? Could the authors please explain what is going on here? Have the authors tried any standard techniques to check if the RcgR behaves as a typical peripheral membrane protein?

As stated in the Materials and Methods section, we enriched cytosolic and membrane protein fractions; however, this procedure does not allow us to compare the relative abundance of RcgR between fractions. Although RcgR lacks predicted transmembrane domains, RcgR::GFP was detected in both cytosolic and membrane fractions. Consistently, microscopy showed RcgR::GFP in the membrane (Fig. 1E, 1F) and in the cytosol (Fig. 1D, 1F), and the fluorescence-intensity quantification (Fig. S2) also supports its presence in both locations.

*** Discussion, lines 381-401, including fig 4. While I do usually appreciate speculation in the discussion section, I find this part a bit too speculative. I think the right panel of Fig 4 and the related discussion text a bit too loose, as the authors do not present any data at all regarding how RcgA could potentially interact with the T4SS. Yes, it is possible that it's interacting with the T4CP, and yes it could also be an integral part of the T4SS, but it could**

also be involved in the initial assembly but not the final functional structure, and several other things. I would suggest that the authors tone down this piece of the discussion.

The reviewer is right in that we might have speculated too much. We thank the reviewer and, as suggested, we have toned down that part of the discussion.

Now Lines 384-404: “As we previously reported, RcgA is essential for pLPU83a transfer. Data obtained in this work suggest that its function is not related to the regulation of the actors involved in the process, since no conjugative genes were differentially expressed in either the proteomic or the transcriptomic analysis of pLPU83aΔrcgA. Both Western blot and microscopy experiments confirmed a membrane localization of RcgA, consistent with previously predicted transmembrane domains (28). Specifically, a continuous localization through the membrane was observed (Fig. 1G-I). A comparable pattern has been reported for the coupling protein (CP) of plasmid R388 (52). Assuming that CP of pLPU83a have a similar distribution pattern and the fact that both proteins are essential for conjugation, a related function or even a direct interaction between RcgA and the CP of pLPU83a is plausible (Fig. 4, 5). In addition, proteins of the vir-T4SS of pTI plasmid from *Agrobacterium tumefaciens* have been localized to multiple foci in a helical or periodic pattern around the circumference of the cells (53). Since deletion of rcgA did not alter transcription of conjugative genes or protein abundance, a structural role in Mpf formation remains possible (Fig. 4, 4). Although the core components of type IV secretion systems (T4SS) in Gram-negative bacteria have been extensively characterized, additional proteins may contribute in the assembly, stabilization, or regulation of these complexes. Supporting this idea, the small membrane protein TraB, encoded by plasmid pIP501 of the Gram-positive bacterium *E. faecalis*, was shown to be essential for conjugative transfer and to interact with multiple mating pair formation proteins, potentially forming part of the complex itself or acting as a recruitment factor (54).”

Minor points

* lines 102-106. There is quite a bit in here that is not greatly explained, like what is LuxR. Further, AHL is stated to be produced by TraI. However, for many readers from the conjugation field TraI is the relaxase in e.g. the F system and pKM101. Would be good to clarify it here to avoid confusion

To avoid confusion, we have now clarified in the text that LuxR is the archetypal AHL-responsive transcriptional regulator in quorum-sensing systems. We also specify that, in the context of rhizobial QS-regulated conjugative plasmids, TraI refers to the AHL synthase (LuxI homolog).

Now, Lines 101-107: “The other well-described regulatory system is based on TraR, a LuxR-like transcriptional regulator. LuxR is the archetypal quorum-sensing (QS)-responsive transcriptional regulator. In known systems, TraR activates conjugative genes in response to acyl-homoserine lactone (AHL) concentrations, a small diffusible molecule produced by TraI (an AHL-synthase in rhizobial plasmids), generally encoded in the same plasmid near traR. As a result, plasmid transfer responds to a QS mechanism, increasing when AHL concentrations are higher (18, 19).”

Re: Spectrum03242-25R1 (**Characterization of RcgA and RcgR, two rhizobial proteins involved in the modulation of plasmid transfer**)

Dear Dr. Gonzalo Arturo Torres Tejerizo:

Your manuscript has been accepted, and I am forwarding it to the ASM production staff for publication. Your paper will first be checked to make sure all elements meet the technical requirements. ASM staff will contact you if anything needs to be revised before copyediting and production can begin. Otherwise, you will be notified when your proofs are ready to be viewed.

Sincerely,
Nicolas Soler
Editor
Microbiology Spectrum

Reviewer #1 (Comments for the Author):

The paper has been carefully revised by the authors. All my issues have been solved.

Reviewer #2 (Comments for the Author):

Thank you for the clarifications in the revision.